# The Role of Cancer-Associated Fibroblasts and Extracellular Vesicles in Tumorigenesis

**DOI:** 10.3390/ijms21186837

**Published:** 2020-09-17

**Authors:** Issraa Shoucair, Fernanda Weber Mello, James Jabalee, Saeideh Maleki, Cathie Garnis

**Affiliations:** 1Department of Integrative Oncology, British Columbia Cancer Research Centre, Vancouver, BC V5Z 1L3, Canada; ishoucair@bccrc.ca (I.S.); fernanda.wmello@gmail.com (F.W.M.); jjabalee@bccrc.ca (J.J.); smaleki@bccrc.ca (S.M.); 2Postgraduate Program in Dentistry, Federal University of Santa Catarina, Florianópolis 88.040-370, Brazil; 3Department of Surgery, University of British Columbia, Vancouver, BC V5Z 1M9, Canada

**Keywords:** extracellular vesicles, tumor microenvironment, cancer-associated fibroblasts, neoplasms

## Abstract

Extracellular vesicles (EVs) play a key role in the communication between cancer cells and stromal components of the tumor microenvironment (TME). In this context, cancer cell-derived EVs can regulate the activation of a CAF phenotype in TME cells, which can be mediated by several EV cargos (e.g., miRNA, proteins, mRNA and lncRNAs). On the other hand, CAF-derived EVs can mediate several processes during tumorigenesis, including tumor growth, invasion, metastasis, and therapy resistance. This review aimed to discuss the molecular aspects of EV-based cross-talk between CAFs and cancer cells during tumorigenesis, in addition to assessing the roles of EV cargo in therapy resistance and pre-metastatic niche formation.

## 1. Introduction

Tumorigenesis is a multi-step process dependent on several modifications at cellular and tissue levels. These modifications lead to sustained proliferative signaling, evasion from growth suppressors and cell death, replicative immortality, and induction of angiogenesis, invasion, and metastasis [1,2]. Malignant tumors are characterized by high cellular heterogeneity, including cancerous and non-cancerous cells, and non-cellular components, composing the tumor microenvironment (TME) [3]. Malignant cells can recruit stromal cells to remodel tissue structure and to secrete growth-promoting stimuli and intermediate metabolites. As a result, stromal components (e.g., fibroblasts, endothelial cells, and pericytes) contribute to multiple processes of tumorigenesis, which include tumor growth, invasion, metastasis, and therapy resistance [3,4].

Cancer-associated fibroblasts (CAFs) are recognized as one of the most abundant stromal cells in the TME. Under the influence of parenchymal cells, stromal cells (including resident fibroblasts, mesenchymal stem cells, and adipocytes) can undergo activation into a CAF-phenotype [5]. CAFs are responsible for remodeling the extracellular matrix (ECM) during tumor progression and metastasis, where they actively participate in proteolysis, crosslinking, and assembly of the ECM, which facilitate malignant cell migration and invasion [6]. Moreover, CAFs also interact with other cells in the TME through direct cell-to-cell contact [7] and paracrine signaling [8].

Extracellular vesicles (EVs) are membrane-enclosed vesicles secreted by most cellular types. These particles play an essential role in the cross-talk between malignant cells and resident cells of the TME [9]. EVs derived from malignant cells can influence the activation of stromal cells into CAFs due to the uptake of EV contents by target cells, including proteins [10] and microRNAs (miRNAs) [11]. Additionally, CAF-derived EVs can regulate malignant and non-malignant cells of the TME, influencing tumor progression [12], metastasis [13] and chemoresistance [14]. This review aims to discuss the role of EVs in the cross-talk between malignant cells and CAFs during tumorigenesis and to understand the roles of EV contents in the process of therapy resistance (including chemotherapy and hormonal therapy) and pre-metastatic niche formation.

## 2. Cancer-Associated Fibroblasts (CAFs)

During homeostatic conditions, resident or quiescent fibroblasts are reported as sentinels of tissue integrity. Showing minimal metabolic and transcriptional activity, they are responsible for ECM structure maintenance and response to mechanical changes by differentiating into myofibroblasts and coordinating tissue repair [15]. In the TME, CAFs enhance tumor growth and invasion by directly affecting tumor cells [5], remodeling the ECM [16,17], inducing angiogenesis and lymphangiogenesis [18,19,20,21], recruiting and promoting the activation of immune cells [22], and supporting cancer-associated inflammation [23]. CAFs exert their pro- or anti-tumorigenic effects by direct cell-to-cell contact or via indirect effects mediated by the secretion of soluble factors or EV contents. While controversies exist in the literature, CAFs have been considered as promising therapeutic targets [24,25,26] and are the subject of ongoing clinical trials that focus on the interference of CAF activation or CAF action in cancer [5].

### 2.1. CAFs Activation

The broad term “CAFs” is used to refer to a heterogeneous population of mesenchymal cells that are responsible for producing several molecules (e.g., cytokines and enzymes), generally defined as the CAF secretome, which can either promote or limit tumor progression [27]. CAFs can originate from both TME resident cells (e.g., fibroblasts, epithelial cells (via EMT), endothelial cells (via endothelial-mesenchymal transition), pericytes (via transdifferentiation), and adipocytes (via transdifferentiation)) and distant cells (e.g., bone marrow-derived mesenchymal stem cells) [28]. This considerable heterogeneity regarding the origin of CAFs also results in great heterogeneity related to the molecular profiles and secretomes of distinct CAF-like cell subpopulations [29]. Some of the most commonly reported molecular markers for positive CAF characterization are α-smooth muscle actin (α-SMA), fibroblast activation protein (FAP), podoplanin, vimentin, and platelet-derived growth factor receptors (PDGFRs); and for negative CAF characterization CD31 and cytokeratin [28,30].

Another important process for CAF differentiation is the metabolic reprogramming, which is linked to several processes, including cancer-cell induced oxidative stress, which modifies mitochondrial function, resulting in higher glucose uptake and reactive oxygen species, culminating in CAF differentiation [31]. Curiously, studies show that CAFs can also undergo autophagy under the influence of the increased demands for energy from other cells of the TME [32]. For instance, cancer cells induce oxidative stress in CAFs, which results in autophagy and secretion of nutrients that enhance an aggressive phenotype in the cancer cells [33]. In this context, cells from the TME secrete high-energy metabolites (e.g., lactate and ketone) and induce metabolic reprogramming in CAFs, including autophagy and aerobic glycolysis, which influence several processes during tumorigenesis, including inflammation, angiogenesis, and tumor growth [34,35].

Several agents can modulate the activation of CAFs in the TME, including inflammation signals, oxidative stress, DNA damage, and transforming growth factor-β (TGF-β) signaling [5]. Among these, the most commonly described mechanisms for CAF profile activation involve TGF-β mediated signaling [36]. For example, TGF-β secreted by breast cancer cells has been reported to induce a CAF phenotype in normal fibroblasts by down-regulating miR-200 and up-regulating transcription factor 12 (TCF12) and friend leukemia virus integration 1 (Fli-1) expression, thereby contributing to ECM remodeling [37]. In colorectal cancer, the C-X-C chemokine receptor type 4 (CXCR4)/TGF-β1 axis has been described as essential for the differentiation of mesenchymal stem cells into CAFs, which in turn promote tumor growth and metastasis [38]. Additionally, long-term exposure to TGF-β has also been reported to induce EMT in epithelial cells, resulting in a myofibroblastic profile via the inactivation of the mitogen-activated protein kinase (MAPK)/extracellular-signal-regulated kinase (ERK) pathway [39].

### 2.2. CAFs Influence Other Components of the TME

During tumorigenesis, CAFs actively participate in collagen turnover and crosslinking, leading to tumor desmoplasia (i.e., excessive collagen deposition in the TME) and increased stiffness [40,41,42]. The role of CAFs in tumor invasion via interactions with the ECM has been reported to involve both matrix metalloproteinase (MMP)-dependent and independent mechanisms. Of note, MMPs are frequently reported to be up-regulated in CAFs in comparison to normal fibroblasts and have been linked to ECM degradation in several types of cancer, promoting cell migration, proliferation, and invasion [43,44,45]. Moreover, there is evidence indicating that the increased secretion of MMPs by CAFs may be linked to drug resistance [46,47]. Studies have also reported that, in addition to proteolysis, CAFs may facilitate basement membrane breaching by cancer cells mechanically, a process that may be related to contractility caused by pulling, stretching, and softening the basement membrane to form gaps [48].

The processes of angiogenesis and lymphangiogenesis can also be enhanced by the CAF secretome in several types of cancer in a multimodal way, mainly involving secretion of soluble factors such as vascular endothelial growth factor (VEGF), platelet-derived growth factor (PDGF), and TGF-β [18,21,49]. CAFs have been reported to promote angiogenesis via VEGF secretion in hepatocellular carcinoma, which is associated with up-regulation of enhancer of zeste homolog 2 (EZH2) and further inhibition of vasohibin 1 (VASH1) [19]. Moreover, the up-regulation of PDGF-C was demonstrated as an alternative mechanism by which tumors presenting resistance to anti-VEGF-A therapy might promote angiogenesis [50]. In addition to soluble signaling factors, CAFs are also able to regulate vascular growth and may contribute to tumor neovascularization by their biomechanical behaviors, which differ from normal fibroblasts [51].

Through immunoediting tumor cells and TME cells, including CAFs, can regulate tumor-promoting and protective mechanisms of the immune system during tumorigenesis. Tumor cells are first eliminated by the immune cells (elimination); then tumor cells and immune cells coexist (equilibrium); and lastly, tumor cell subclones modify their immunogenicity and evade immune recognition (escape), promoting tumor progression [22]. In this context, CAFs can modulate the tumor immune response (i.e., induction of immunosuppression) to facilitate tumor progression, including interactions with several immune cells, such as macrophages, T cells, dendritic cells, and myeloid cells. For instance, CAFs can recruit monocytes by secreting monocyte chemotactic protein-1 (MCP-1) and stromal cell-derived factor-1 (SDF-1); they are also able to induce transdifferentiation of M1 macrophages to M2 macrophages, which results in immunosuppression and increased cancer cell proliferation [52]. Additionally, CAFs can indirectly reduce T cells activation by secreting cytokines, such as TGF-β, that modulate antigen presentation, which is primarily associated with CAFs effects in dendritic cells [53].

Interestingly, recent evidence suggests that a specific CAF sub-population—namely, inflammatory CAFs (iCAFs)—is responsible for inducing and maintaining an inflammatory microenvironment through the secretion of several pro-inflammatory cytokines (e.g., IL-6 and CXCL1) [54,55]. Inflammatory cytokines and chemokines play different roles in tumor progression, including EMT promotion, invasion, and metastasis [56,57]. Therefore, through the modulation of tumor-associated inflammation, iCAFs also interfere in several processes of tumorigenesis. For example, CAF-derived IL-1β can induce C motif chemokine ligand 22 (CCL22) mRNA expression in oral cancer cell lines through the activation of transcription factor nuclear factor kappa B (NF-κB), which is associated with cell transformation and regulatory T cell infiltration [58]. The interactions between CAFs and the immune and inflammatory network is complex and includes several types of cells (e.g., cytotoxic and helper T cells, macrophages, neutrophils, dendritic cells, natural killer cells, and myeloid-derived suppressor cells) and related cytokines, chemokines, and soluble factors (e.g., ILs, TGF-β, and TNF-α). While outside of the main scope of the present review, these interactions between CAF and the immune system and inflammation have been previously reviewed [22,23].

Direct interactions between CAFs and cancer cells via soluble factors and EV-mediated paracrine signaling or cell-to-cell communications are vastly explored in literature. Studies show that CAFs can influence processes involved in virtually all major hallmarks of cancer [30,49,59]. The effects of soluble factors’ signaling and cell-cell communications in the cross-talk between cancer cells and CAFs were previously reviewed by other researchers [5,60,61], and therefore, will not be further discussed. Interactions between cancer cells and CAFs mediated by EV cargo are detailed Section 4 and Section 5 of this review.

## 3. Extracellular Vesicles (EVs)

EVs are a class of phospholipid-bilayer-enclosed membranes carrying a variety of biological molecules, including nucleic acids, proteins, and lipids. EVs are released by virtually all cell types in an evolutionarily conserved manner across eukaryotes and prokaryotes [62,63]. EVs can act on local and distant sites and can be found circulating in a wide range of biological fluids, including blood, urine, bile, nasal secretions, and saliva [64]. EVs can be broadly assigned to three main categories based on origin: exosomes, microvesicles (MVs), and apoptotic bodies.

Exosomes (30–150 nm) are formed through endosomal trafficking, beginning as late endosomes that mature into multivesicular bodies (MVBs). Intervesicular bodies (ILVs) are formed through the inward budding of the limiting membrane of MVBs [62,65]. The most well-characterized pathway for ILV biogenesis is through an endosomal sorting complex required for transport (ESCRT) complex-dependent mechanism. ESCRT (-0, -I, -II) recognizes and sequesters ubiquitinated proteins on the endosomal membrane, followed by inward budding and scission by ESCRT-III [63,66]. Cargo clustering may also occur through syntenin along with the ESCRT accessory protein ALG-2-interacting protein X (ALIX), with only ESCRT-III being required for ILV biogenesis [67,68]. ILV formation has also been shown to occur through ESRT-independent mechanisms via tetraspanins (CD63) [62,69], and a lipid mechanism involving ceramides made by sphingomyelinases (nSMases) [62,70]. MVBs may be trafficked to the lysosome for degradation, or they may be trafficked by guanosine triphosphatases (GTPases) Rab27a and Rab27b towards the plasma membrane for docking and fusion [62]. After the MVB fuses with the plasma membrane, its ILVs will be released into the extracellular space as exosomes [63] (Figure 1).

Microvesicles (50–1000 nm) are a heterogeneous group that includes EVs such as oncosomes, migrasomes, and arrestin domain-containing protein 1 (ARRDC1)-mediated microvesicles (ARMMs) [62,71]. MVs are formed through the outward budding of the plasma membrane (Figure 1). Oncosomes are large EVs associated with cancer. Migrasomes are involved in the transportation of multivesicular cytoplasmic content during cell migration [62]. Additionally, apoptotic bodies (500–4000 nm) result from the disintegration of the plasma membrane of apoptotic cells [67,72].

Once released into the extracellular milieu, EV-mediated communication may act in an autocrine or paracrine manner. Once at the recipient cells, EVs may remain at the plasma membrane, fuse with the plasma membrane, or be internalized by the target cell. Endocytosis of EVs may be clathrin-mediated or clathrin-independent (macropinocytosis or phagocytosis), or facilitated via caveolae or lipid rafts [62]. Intracellular communication via EVs occurs during homeostasis and under pathological conditions, including cancer. The cargo carried by EVs is selectively rather than indiscriminately packaged. This suggests a biological role for the contents that are packaged, since selection can be driven to create conditions that either promote or inhibit malignancy [73].

There are several techniques employed currently to isolate EVs. Differential ultracentrifugation (DU) is widely used as a low-cost, high-throughput method to isolate EVs from large sample sizes. DU is often combined with other purification techniques such as ultrafiltration or density gradient centrifugation to yield increased particle purity. Size-exclusion chromatography yields EV pellets of high purity but dilutes samples, which need to then be re-concentrated. Immuno-affinity and microfluidics are techniques based on EV characteristics such as surface markers (e.g., CD63) but there is currently no marker that can accurately discern between various EV subtypes. Polymer-based precipitation captures EVs in polymer nets based on size using simple centrifugation, making it useful for clinical usage. EV samples are further characterized commonly through transmission electron microscopy (TEM), nanoparticle tracking analysis, Western blots for common markers (TSG101, Alix, CD63, CD81, Floatillin), and flow-cytometry [74,75].

Damaged and diseased cells, including cancer, have been shown to shed higher amounts of EVs compared to their healthy counterparts. This heightened production may be linked to the extensive metabolic reprogramming that cancer cells undergo [76]. The mechanisms regulating EV production are unclear; however, EV formation and release have been inhibited by targeting biological molecules involved in EV trafficking (calpeptin, manumycin A, Y27632) or lipid metabolism (D-pantethine, imipramine, GW4869) [77]. In regard to fibroblasts, GW4869 which targets nSMases to inhibit exosome generation and release, has been extensively used in CAF-EV research [13,78,79,80,81,82,83].

Several studies [79,81,84,85,86,87,88,89,90] have included an investigation into clinical translations of their findings, wherein CAF-EV cargo expression in tissue or blood based biopsies was found to be linked to clinical features such as overall survival [88], lymph node metastasis [87], metastasis [84], and poor prognosis [79].

## 4. Role of EV Contents in CAF Activation

The transfer of molecular cargo from cancer to stromal cells via EVs is a key regulator of CAF differentiation. Stromal cells, particularly fibroblasts, take up large numbers of cancer cell-derived EVs when compared to epithelial cells, making them important targets of EV-mediated cross-talk [91]. Additionally, endothelial cells, pericytes, and mesenchymal stem cells can also be induced to a CAF phenotype by cancer-derived EVs [92,93,94,95,96,97]. The ability of cancer-derived EVs to promote CAF phenotype has been linked to several types of molecular cargo, including miRNAs, proteins, and to a lesser extent, messenger RNAs (mRNAs) and long non-coding RNAs (lncRNAs) (Table 1 and Figure 2).

### 4.1. miRNA

MiRNAs are small (≈18–22 nucleotides) non-coding RNAs that regulate gene expression at a post-transcriptional level. MiRNAs are among the most abundant cargo found in EVs; as such, it is unsurprising that EV-mediated miRNA transfer from cancer to stromal cells plays an important role in promoting CAF differentiation. To date, at least two dozen miRNAs have been found to be responsible for promoting EV-mediated CAF differentiation in a variety of cancer types [11,95,98,99,100,101,102,103,104,105,106,107,108,109,110,111,112,113,114,115,116]. Uptake of miRNA-containing EVs into recipient cells results in altered expression of key signaling pathways, many of which have a known oncogenic or tumor-suppressive role, which leads to the adoption of a CAF phenotype. For instance, miR-21 has been found to be highly expressed in EVs derived from various cancer types, including multiple myeloma and oral, colon, liver, and prostate cancer [95,99,103,104,114]. Uptake of miR-21-containing EVs into recipient fibroblasts actively promotes CAF differentiation via an increase in PI3K signaling [99,104]. The same study found that CAF differentiation correlated with increased expression of additional oncogenic proteins in recipient cells, including β-catenin, signal transducer and activator of transcription 3 (STAT3), the mammalian target of rapamycin (mTOR), and TGF-β [99]. STAT3 signaling was found to be activated by miR-155 and miR-210, which are expressed at high levels in EVs of several cancer types, the uptake of which by recipient fibroblasts promotes CAF differentiation [98,105]. STAT3 signaling has long been associated with angiogenesis, and in agreement, STAT3 signaling in fibroblasts was found to drive the expression of the pro-angiogenic factors VEGF, MMP-9, and fibroblast growth factor 2 (FGF2), prompting CAFs to adopt a pro-angiogenic phenotype [98]. Although miR-155 and miR-210 both promote a STAT3-dependent pro-angiogenic phenotype in recipient fibroblasts, they do so by impacting different intracellular targets (suppressor of cytokine signaling 1 (SOCS1) and tet methylcytosine dioxygenase 2 (TET2), respectively). This observation demonstrates how tumor cells can achieve similar outcomes via different mechanisms.

In addition to the activation of oncogenic pathways, several studies have shown that EVs miRNAs can induce a CAF phenotype in recipient cells by inhibiting the tumor suppressor p53 and related proteins [101,102,115]. High expressions of miR-125b, miR-155, miR-1249-5p, miR-6737-5p, and miR-6819-5p in cancer cell-derived EVs have been shown to drive CAF differentiation in recipient fibroblasts via inhibition of the p53 pathway [101,102,115]. Interestingly, loss of p53 signaling in colon cancer cells resulted in increased expression of EV microRNAs that target p53 in recipient cells [101]. This result suggests that the onset of CAF differentiation in many tumors may correlate with p53 inhibition in cancer cells, a common and early event in tumorigenesis. Whereas inhibition of p53 in fibroblasts promotes a CAF phenotype, inhibition of another tumor suppressor, breast cancer type 1 susceptibility protein (BRCA1), allows cancer cell-derived EVs to promote an oncogenic phenotype in recipient fibroblasts that is similar to that of the inducing cancer cells [117,118,119]. This oncogenic phenotype is characterized by mesenchymal-to-epithelial transition, deregulation of gene and miRNA expression that mimics that of the parental tumor, and the ability to form solid tumors in mice [117,118,119]. Further investigation is necessary to determine why the inhibition of these tumor suppressors results in different outcomes.

**Table 1 ijms-21-06837-t001:** Summary of effects of cancer cell-derived extracellular vesicles on stromal cells.

Author (Year)	Type of Cancer	EV Type; Size *	EV Isolation Method	Target Cargo	Main Results
**MicroRNA**					
Chen (2019) [99]	OSCC	EVs; size NR	Precipitation via Total Exosome Isolation Reagent Kit	miR-21	Oral cancer cell-derived EVs are enriched for miR-21-5p, and this enriched is reduced by treatment with ovatodiolide. Loss of miR-21-5p correlated with reduced tumorigenesis in vitro and in vivo.
Zhou (2018) [104]	Hepatocellular carcinoma	sEVs (exosomes); mean range 70.6–78.4 nm	Ultracentrifugation with filtration	miR-21	Hepatocellular carcinoma cell-derived EVs contain high levels of miR-21 that promotes CAF differentiation in recipient hepatocyte stellate cells by inhibiting PTEN and activating PDK1/AKT signaling.
Ju (2019) [103]	Colon	EVs; 40–400 nm	Precipitation via ExoQuick-TC	miR-21-3p; miR-769-3p	Colorectal cancer cells with a gain-of-function mutation in *TP53* secrete high levels of miR-21-3p and miR-769-3p in their exosomes. These microRNAs activate recipient fibroblasts via the TGF-β/SMAD pathway.
Cheng (2017) [95]	Multiple myeloma	EVs; 30–310 nm	Ultracentrifugation	miR-21; miR-146a	Multiple myeloma cells release exosomes high in miR-21 and miR-146a that promote CAF differentiation, proliferation, and IL-6 release from MSCs.
Sanchez (2015) [114]	Prostate	EVs; size NR	Precipitation via ExoQuick-TC	miR-21-5p; miR-100-5p; miR-139-5p	Prostate cancer cell-derived EVs contain high levels of miR-100-5p, miR-21-5p, and miR-139-5p that, when taken up by recipient fibroblasts, promote migration and secretion of MMPs.
Zhou (2018) [105]	Melanoma	EVs; 50–200 nm	Ultracentrifugation	miR-155-5p	MiR-155-containing, melanoma-derived EVs are taken up by CAFs, leading to suppressed SOCS1, increased JAK/STAT signaling and expression of pro-angiogenic factors.
Pang (2015) [115]	Pancreas	EVs (microvesicles); size NR	Ultracentrifugation	miR-155	Prostate cancer cell-derived microvesicles contain miR-155 that, when taken up by recipient fibroblasts, promotes a CAF-like phenotype via down-regulation of *TP53INP1*.
Shu (2018) [108]	Melanoma	sEVs (exosomes); 51–63.7 nm	Ultracentrifugation	miR-155; miR-210	miR-155 and miR-210 released in the EVs of melanoma cells are taken up by recipient fibroblasts, resulting in increased glycolysis and decreased oxidative phosphorylation.
Fan (2020) [98]	Lung	EVs; 50–200 nm	Ultracentrifugation	miR-210	MiR-210 in lung cancer cell-derived EVs is transferred to fibroblasts, where it inhibits TET expression, leading to CAF differentiation, activation of JAK/STAT signaling, and expression of pro-angiogenic VEGF, MMP9, and FGF2.
Wang (2018) [107]	Gastric	EVs; size NR	Precipitation via ExoQuick-TC	miR-27a	miR-27a is expressed at high levels in the EVs of gastric cancer cell-derived EVs and inhibits CSRP2 in recipient fibroblasts, resulting in CAF differentiation, and increased proliferation and migration.
Lucchetti (2017) [112]	Colon	sEVs (exosomes); 30–100 nm	Ultracentrifugation	miR-24; miR-26a; miR-27a	Treatment of colon cancer cells with sodium butyrate increased the levels of various microRNAs in EVs. Such EVs induced proliferation and migration of recipient fibroblasts.
Abdouh (2019) [117]	Colorectal	sEVs (exosomes); 50–120 nm	Ultracentrifugation	Various microRNAs	Colon cancer cell-derived EVs transfer various microRNAs to recipient *BRCA1*-knockout fibroblasts and induce a malignant phenotype.
Zhang (2019) [100]	Ovarian	sEVs (exosomes); mode 92.6 nm	Precipitation via ExoQuick-TC	miR-124	EVs secreted by normal ovarian cells, but not ovarian cancer cells, deliver miR-124 to fibroblasts and inhibit their differentiation into CAFs.
Yoshii (2019) [101]	Colon	sEVs (exosomes); mean 100 nm	Ultracentrifugation	miR-1249-5p; miR-6737-5p; miR-6819-5p	*TP53*-deficient colon cancer cells secrete EVs containing microRNAs that inhibit p53 in recipient fibroblasts and drive their differentiation into CAFs.
Vu (2019) [102]	Breast	EVs; 50–300 nm	Ultracentrifugation	miR-125b	Breast cancer cells secrete EVs containing high levels of miR-125b that drives CAF differentiation in recipient fibroblasts by inhibiting TP53INP1 and TP53.
Dai (2018) [11]	Colorectal	EVs; size NR	Ultracentrifugation	miR-10b	Uptake of miR-10b-containing EVs by fibroblasts inhibits *PIK3CA* expression, increases TGF-β and α-SMA, and reduces proliferation.
Yan (2018) [106]	Breast	EVs; size NR	Ultracentrifugation	miR-105	Oncogenic MYC expression in breast cancer cells induces expression and EV packaging of miR-105 that, when taken up by recipient fibroblasts, induces a metabolic shift. Reprogrammed fibroblasts increase glucose and glutamine metabolism and detoxify metabolic wastes (lactate, ammonium) and convert them into energy-rich metabolites for cancer cells.
Lawson (2018) [109]	Lung	EVs; size NR	Ultracentrifugation	miR-142	Lung cancer cells secrete EVs high in miR-142 that induces a CAF phenotype in recipient cells.
Gong (2018) [110]	Osteosarcoma	sEVs (exosomes); 100–120 nm	Ultracentrifugation	miR-675	Metastatic osteosarcoma cells secrete exosomes containing high levels of miR-675 that induces migration and invasion in recipient fibroblasts by down-regulating CALN1.
Fang (2018) [111]	Hepatocellular carcinoma	EVs; 40–370 nm	Ultracentrifugation	miR-1247-3p	Metastatic hepatocellular carcinoma cells secrete exosomes containing high levels of miR-1247-3p that, when taken up by recipient fibroblasts, inhibits B4GALT3 and activates β1-integrin-NF-κB signaling. Activated fibroblasts secrete inflammatory cytokines.
Baroni (2016) [113]	Breast	EVs; size NR	Precipitation via ExoQuick-TC	miR-9	Breast cancer cell-derived EVs contain high levels of miR-9 that promote migration, invasion, and CAF differentiation in recipient fibroblasts. In turn, fibroblasts secrete miR-9 in EVs that can inhibit E-cadherin in epithelial cells.
Morello (2013) [116]	Prostate	Large oncosomes; 1–10 µm	Ultracentrifugation	miR-1227	Large oncosomes released by pancreatic cancer cells contain high levels of miR-1227 that promote CAF migration.
**Protein**					
Ringuette Goulet (2018) [10]	Bladder	EVs; 30–450 nm	Precipitation by Total Exosome Precipitation Reagent	TGF-β	Bladder cancer cell-derived exosomes trigger differentiation of recipient fibroblast to CAFs. Cancer-cell exosomal TGF-β localized within the exosomes is released to bind surface TGF-β receptors of fibroblasts.
Yeon (2018) [92]	Melanoma	EVs; 30–350 nm	Ultracentrifugation	TGF-β	Melanoma-derived exosomes promote endothelial-to-mesenchymal transition in recipient HUVECs and CAF differentiation, whereas MSC-derived exosomes suppressed this transition.
Wei (2017) [120]	Peritoneal	sEVs (exosomes); 30–100 nm	Precipitation via ExoQuick Exosome Precipitation Solution	TGF-β	Ascites-derived exosomes promote proliferation and migration of recipient mesothelial-mesenchymal cells and trigger differentiation to CAFs.
Webber (2015) [121]	Prostate	EVs; mode 115 nm	Ultracentrifugation with sucrose cushion	TGF-β	Exosomal TGF-β is necessary for stromal differentiation to a CAF phenotype.
Chowdhury (2014) [96]	Prostate	EVs; size NR	Ultracentrifugation with filtration and sucrose/D_2_O cushion	TGF-β	Cancer cell-derived exosomes promote bone marrow mesenchymal stem cell differentiation into myofibroblasts; these CAFs promote angiogenesis, and tumor cell proliferation, migration, and invasion.
Gu (2012) [97]	Gastric	sEVs (exosomes); 40–100 nm	Ultrafiltration with a 100 kDa MWCO hollow membrane; ultracentrifugation with a sucrose cushion	TGF-β	Cancer cell-derived exosomal TGF-β activates TGF-β/SMAD signaling in recipient mesenchymal stem cells and triggers their differentiation to CAFs.
Webber (2010) [122]	Mesothelioma; Prostate; Bladder; Colorectal; Breast	sEVs (exosomes); size NR	Ultracentrifugation with a 30% sucrose/D_2_O cushion	TGF-β	Cancer cell-derived exosomes bearing TGF-β on their surface elicit differentiation of recipient fibroblasts into myofibroblasts; this ability may be attenuated through loss of exosomal betaglycan.
Chen (2019) [99]	Oral	EVs; size NR	Precipitation via Total Exosome Isolation Reagent	TGF-β; STAT3; mTOR	Oral CSC-derived exosomes promoted cisplatin resistance and CAF differentiation. Ovatodiolide treatment suppressed the pro-tumorigenic effects of exosomes.
Aoki (2017) [123]	Epitheloid sarcoma	EVs (microvesicles); 100–300 nm	Differential centrifugation (50,000 g for 1 h)	CD147	Epitheloid sarcoma cells shed microvesicles high in CD147 that promotes MMP2 expression in recipient fibroblasts.
Hatanaka (2014) [124]	Melanoma	sEVs (exosomes); mean 100 nm EVs (microvesicles); 100–800 nm	Ultracentrifugation	CD147	CD147-containing microvesicles shed from malignant melanoma cells promote MMP2 expression in recipient fibroblasts.
Zhang (2013) [125]	Hepatocellular carcinoma	EVs (microvesicles); 200–500 nm	Ultracentrifugation	CD147	ANXA2 promotes shedding of CD147-containing microvesicles from hepatocellular carcinoma cells. CD147 promotes MMP2 expression in recipient fibroblasts.
Sung (2020) [126]	Breast	EVs; size NR	Precipitation via ExoQuick-TC	ITGB4	BNIP3L-dependent mitophagy and reverse Warburg effect was induced in CAFs through the uptake of cancer cell-derived exosomal ITGB4.
Wu (2020) [127]	Nasopharyngeal carcinoma	EVs; 30–250 nm	Precipitation via ExoQuick-TC	LMP1	Activation of the NF-kB p65 pathway through transfer of LMP1 to normal fibroblasts induces CAF activation, a reverse Warburg effect; EV LMP1 promoted proliferation, migration, and radiation resistance in tumor cells.
Zhang (2019) [128]	Pancreas	EVs; majority <100 nm	Differential centrifugation with filtration	Lin28B	EVs secreted by pancreatic cancer cells transfer Lin28B to recipient cancer cells. Uptake of Lin28B increases expression of PDGFB that promotes recruitment of pancreatic stellate cells to the premetastatic niche.
Frassanito (2019) [129]	Multiple myeloma	EVs; size NR	Differential centrifugation with filtration (0.22 μm); precipitation via ExoQuick-TC	WWC2	EVs secreted by myeloma cells contain WWC2, a regulator of the Hippo pathway. WWC2 in fibroblasts promotes their transition to CAFs and up-regulates their expression of miR-27b-3p and miR-214-3p, which protect CAFs from apoptosis.
Urciuoli (2018) [130]	Osteosarcoma	EVs; mean 150–175 nm	Precipitation via ExoQuick-TC	MMP-9; TNF- α	Osteosarcoma-derived EVs increase fibroblast proliferation and induce a tumor-like phenotype.
El Buri (2018) [131]	Breast	EVs; size NR	Ultracentrifugation	S1PR2	Breast cancer cells secrete exosomes containing S1PR2 that if processed to a shorter form that activates ERK1/2 signaling and proliferation in recipient fibroblasts.
McAtee (2018) [132]	Prostate	sEVs (exosomes); mean 112 nm	Ultracentrifugation	Hyal1	Prostate cancer cell-derived exosomes contain Hyal1, a hyaluronidase that is transferred to recipient stromal cells. Uptake of Hyal1-positive exosomes greatly increased stromal cell motility, enhanced adhesion to type IV collagen, and increased FAK phosphorylation and integrin engagement.
Overmiller (2017) [133]	Squamous cell carcinoma	EVs; mode 134 nm (ultracentrifugation), mode 127 nm (polymer)	Culture media: ultracentrifugation with filtration or precipitation via ExoQuick; Serum: ExoQuick	EGFR; DSG2 C-terminal fragment	DSG2 over-expression increased EV release. Cancer cell-derived EVs activated Erk1/2 and Akt pathways, and promoted proliferation in recipient fibroblasts. DSG2 C-terminal fragment and EGFR were up-regulated in HNSCC patient serum EVs.
Silva (2016) [134]	Mammary carcinoma	EVs; 10–690 nm	Ultracentrifugation with filtration	AHNAK	Cancer cell-derived EVs enriched in AHNAK promote migration in recipient fibroblasts; AHNAK inhibition had no significant effect on cancer cell proliferation or viability, but did repress vesicle production.
Kreger (2016) [135]	Breast	sEVs (exosomes); mean 30–40 nm EVs (microvesicles); size NR	sEVs: Ultracentrifugation with filtration; Microvesicles: particles captured by 0.22 μm filter	Survivin	Treatment of metastatic breast cancer cells with paclitaxel or nocodazole, but not drugs with other mechanisms of action, causes the cells to release exosomes with high levels of Survivin. These Survivin-containing exosomes promote survival of serum-starved fibroblasts and cancer cells.
Zhao (2015) [136]	Melanoma	EVs (microvesicles); 100–1000 nm	Differential centrifugation (14,000 g for 2 h)	ERK1/2	Fibroblasts exposed to tumor-derived microvesicles increased expression of VCAM-1; this up-regulation could be repressed through ERK1/2 inhibition via U0126; tumor-derived microvesicles had no significant impact on fibroblast proliferation or apoptosis but did promote CAF differentiation.
**Other cargo (mRNA, lncRNA)**					
Hu (2019) [137]	Melanoma	sEVs (exosomes); mean 127–132 nm	Ultracentrifugation with filtration	Gm26809	Exosomes secreted by melanoma cells induced CAF activation in embryotic fibroblasts and increased cell migration.
Gener Lahav (2019) [138]	Melanoma	sEVs (exosomes); 30–100 nm	Ultracentrifugation with filtration	Non-specific	Melanoma-derived EVs induced pro-inflammatory signaling in lung fibroblasts and astrocytes.
Uriciuoli (2018) [130]	Osteossarcoma	EVs; mean 150–175 nm	Precipitation via ExoQuick	Non-specific	Treatment of fibroblasts with osteosarcoma-derived EVs induced substantial biological and functional effects in recipient cells.
Gutkin (2016) [139]	T cell leukemia; Breast; Chronic myeloid leukemia; Colon	sEVs (exosomes); 30–10 nm	Precipitation miRCURY Exosome Isolation Kit	hTERT mRNA	Uptake of EVs containing hTERT mRNA by fibroblasts resulted in increased telomerase activity and phenotypic changes, including increased proliferation, extension of life span and postponement of senescence.

Legend: (*) Size is reported as range unless otherwise stated. α-SMA, α smooth muscle actin; AKT, RAC-alpha serine/threonine protein kinase; ANXA2; annexin A2; B4GALT3, beta-1,4-galactosyltransferase 3; BNIP3L, BCL2 interacting protein-like 3; BRCA1, breast cancer associated 1; CAF, cancer-associated fibroblast; CALN1, calneuron 1; CSC, cancer stem cell; CD147, cluster of differentiation 147; CSRP2, cysteine and glycine-rich protein 2; DSG2, desmoglein 2; EGFR, epidermal growth factor receptor; ERK, extracellular signal-regulated protein kinase; EV, extracellular vesicle; FGF, fibroblast growth factor; HNSCC, head and neck squamous cell carcinoma; HUVEC, human umbilical vein endothelial cells; Hyal1, hyaluronidase 1; IL-6, interleukin 6; ITGB4, integrin subunit beta 4; JAK/STAT, Janus kinase/Signal transducer and activator of transcription; LMP1, latent membrane protein 1; MSC, mesenchymal stem cell; MMP, matrix metalloproteinase; mTOR, mammalian target of rapamycin; NF-κB, nuclear factor kappa-light-chain-enhancer of activated B cells; NR, not reported; OSCC, oral squamous cell carcinoma; PDK1, phosphoinositide-dependent kinase 1; PIK3CA, phosphatidylinositol-4,5-bisphosphate 3-kinase catalytic subunit alpha; PTEN, phosphatase and tensin homolog; S1PR2, sphingosine-1-phosphate receptor 2; sEVs, small extracellular vesicles; SOCS1, suppressor of cytokine signaling 1; TET, ten-eleven translocation; TGF-β, transforming growth factor β; TNF-α, tumor necrosis factor α; TP53; tumor protein 53; TP53INP1, tumor protein 53 inducible nuclear protein 1; VEGF, vascular endothelial growth factor; WWC2, WW and C2 domain-containing 2.

### 4.2. Proteins

Whereas EV miRNAs function by post-transcriptional regulation of gene expression in recipient cells, EV proteins are thought to stimulate receptors at the recipient cell surface or to become active in the recipient cell upon EV uptake. Indeed, given the high degree of overlap between pathways targeted by EV miRNAs and proteins, these and other cargo types may work together to promote a CAF phenotype as described above for STAT3. Another example of EV cargo cooperation is provided by alterations of the TGF-β/SMAD signaling pathway. Multiple cargos, including miR-21, miR-769-3p, TGF-β protein, and TGF-β mRNA have been found to drive a CAF phenotype in recipient cells via TGF-β signaling [10,92,96,97,99,103,120,121,122,130]. These cargos can also coexist within the same EVs [99]. Vesicle-bound TGF-β may be tethered to the EV membrane or may exist within the soluble compartment of the EV; the preferred form likely depends upon the characteristics of the parental cell [10,92,122]. Regardless of whether TGF-β is located on the EV surface or in the EV lumen, EV-bound TGF-β is a potent stimulator of SMAD signaling, which drives expression of genes required for CAF differentiation, including the known CAF marker α-SMA [10,97,122]. In turn, newly differentiated CAFs secrete high levels of TGF-β that promote a CAF phenotype in nearby cells and can promote increased metastasis of tumor cells, thus driving enhanced tumorigenesis [103].

In addition to TGF-β, several other EV proteins have been found to regulate CAF differentiation, including the integrin very late antigen-4 (VLA-4) and the virally-encoded oncoprotein latent membrane protein 1 (LMP1) [127,136]. Specifically, VLA-4 has been reported to be expressed at high levels in EVs obtained from melanoma cells and has been reported to trigger ERK1/2 signaling in recipient cells, which results in up-regulation of CAF markers and inflammatory factors [136]. Interestingly, oncogenic LMP1, encoded by the Epstein–Barr Virus (EBV), has been found to be incorporated into the EVs of virally-infected cells and transferred to recipient fibroblasts, resulting in induction of a CAF phenotype via activation of NF-κB signaling [127]. Further, LMP1 uptake has been reported to promote aerobic glycolysis in CAFs. This allows fibroblasts to release high levels of lactate and other glycolytic intermediates that promote tumorigenesis by providing tumor cells with nutrients and by acidifying the TME, a mechanism known as the reverse Warburg effect [127]. These results demonstrate how EV-mediated transfer of a single protein can have a profound effect on fibroblast differentiation and function.

Several additional proteins have been described that alter CAF function without necessarily driving CAF differentiation. For instance, the proteins desmoglein 2 and AHNAK have been found to regulate the release and packaging of EVs in head and neck cancer and breast cancer, respectively [133,134]. The knockdown of these proteins results in a decrease in EV release and the ability of EVs to promote recipient cell proliferation and migration [133,134]. Still, other EV proteins have been found to promote various CAF functions, including aerobic glycolysis (integrin subunit beta 4 (ITGB4)) and CAF recruitment to metastatic tumors (Lin-28 homolog B (LIN28B)), resistance to apoptosis (WW and C2 domain containing 2 (WWC2) and survivin), increased migration or proliferation (hyaluronidase-1 (Hyal1), sphingosine-1-phosphate receptor 2 (S1PR2)), and expression of MMPs (CD147) [123,124,125,126,128,129,131,132,135]. These results demonstrate the wide range of effects that cancer cell-derived EVs have on cells of the TME.

### 4.3. Other Cargos (mRNA and lncRNA)

Several studies demonstrate roles in tumorigenesis for other EV cargo types, including mRNA and lncRNA. Indeed, melanoma cell-derived EVs exhibited differential expression of over 1600 transcripts when compared to fibroblast-derived EVs, suggesting an important role for mRNA incorporation into EVs [138]. Specific mRNAs, including those encoding TNF-α, TGF-β, and IL-6, have been found to be transferred from cancer cells to fibroblasts and to regulate fibroblast differentiation and function [130]. Interestingly, various cancer cell lines have been found to package human telomerase reverse transcriptase (hTERT) mRNA into their EVs; uptake of this cargo by recipient fibroblasts results in telomerase-positive fibroblasts that exhibit increased proliferation, protection from late senescence, and protection from DNA damage [139]. Additionally, the lncRNA Gm26809 was found in melanoma-derived EVs and induced expression of CAF markers in recipient cells [137]. Although reports of mRNA and lncRNA transfer remain sparse, current data suggest an important role for these cargo types in EV-mediated CAF induction.

## 5. CAFs EVs Contents in the Communication with Malignant Cells

The dynamic network established between cancer cells and CAFs plays an essential role in tumor development and progression. As previously mentioned, cancer cells may induce a CAF phenotype on other cells from the TME through EV-mediated signaling. These CAFs may, in turn, release EVs of their own to transfer cargo and stimulate cancer cells. Numerous studies have investigated the role of CAF-derived EVs and the influence of their contents on recipient cancer cells. The influence of specific CAF-derived EV cargo on cancer cells has been described in a wide range of cancers (Table 2) [12,13,78,79,80,81,82,83,84,85,86,87,88,89,90,140,141,142,143,144,145,146,147,148,149,150,151,152,153,154,155,156]. The ability of CAF-derived EVs to promote tumorigenic effects in recipient cancer cells has been linked to proteins and non-coding RNAs, including miRNAs and lncRNAs. The impact of CAF-derived EV cargo has a range of effects, including the promotion of EMT, invasion, metastasis, stemness, proliferation, growth, chemoresistance, and metabolic reprogramming, and the inhibition of apoptosis (Figure 2).

There is a lack of evidence supporting any anti-tumor effects by CAF-EVs specifically; however, some studies have shown possible anti-tumor effects by CAFs [157]. For instance, one study targeted the extensive fibrosis and stromal myofibroblasts associated with pancreatic ductal carcinomas. The deletion of α-SMA myofibroblasts in pancreatic ductal carcinomas mice models enhanced hypoxia, EMT, CSCs, and chemoresistance, and reduced animal survival. A decrease in immune surveillance and increased Treg infiltration were also observed, leading the authors to suggest that the fibrosis associated with pancreatic ductal carcinomas is part of a host immune response [158].

### 5.1. miRNA

Most studies examined the transfer of EV miRNAs from CAFs to cancer cells [12,79,81,83,84,85,86,88,89,142,143,144,148,150,152,154,155,156]. A miRNA that has been reported across multiple studies is miR-21, which is enriched in CAF-derived EVs in breast, ovarian, and colorectal cancers [12,143,150]. In these studies, mir-21 acted as an oncogenic miRNA (oncomiR), by promoting tumorigenesis in cancer cells. In other instances, it was the repression of specific miRNAs in CAF-derived EVs that conferred pro-tumorigenic behaviors in recipient cells. For example, in head and neck cancer (HNC), miR-3188 acts as a tumor suppressor by inhibiting cell proliferation and promoting apoptosis in recipient cancer cells. The loss of miR-3188 in CAF-derived exosomes promotes tumor progression through the de-repression of B-cell lymphoma 2 (BCL2). Clinically, this has been reflected in the observed low levels of circulating miR-3188 in plasma samples from HNC patients [85].

### 5.2. Proteins

Several studies have observed that the protein contents of CAF-derived EVs are able to induce changes in recipient cancer cells, including increased migration, invasion, EMT, and metastasis [13,80,84,87,90,140,141,145,146,149,151,153]. Specific protein cargo has been shown to stimulate pro-tumor activity but can also regulate the uptake of CAF-derived EVs by recipient cancer cells. In pancreatic cancer, cell aggressiveness characterized by an increase in proliferation and metastasis is dependent on the uptake of annexin A6 (ANXA6) positive CAF-derived exosomes. Loss of ANXA6 has been reported to destabilize the ANXA6/low-density lipoprotein receptor-related protein 1 (LRP1)/thrombospondin-1 (TSP1) complex, which leads to reduced uptake of CAF-derived exosomes by cancer cells, and a reduction in their proliferative and migratory abilities. ANXA6 enrichment in CAF-derived exosomes is reflected in the serum-derived exosomes of pancreatic ductal adenocarcinoma patients, where elevated ANXA6 levels are associated with higher tumor grades and poor clinical outcomes [90].

In addition to the transfer of protein, CAF-derived EVs may carry protein markers that influence their uptake by specific cancer cell types. One study found that CAF-derived exosomes are taken up by scirrhous-type gastric cancer cells but not poorly differentiated or intestinal-type gastric cancer cells. These exosomes are positive for the surface marker CD9 when secreted from CAFs, but not when secreted by normal fibroblasts. Interestingly, these CAF-derived exosomes are negative for other common exosome surface markers, CD63 and CD81. CD9+ CAF-derived exosomes stimulate migration and invasion in scirrhous-type gastric cancer cells, and promote increased MMP-2 activity [87]. In line with this study, other studies corroborate the ability of CAFs to promote the degradation of the ECM through up-regulation of metalloproteinases. CAFs can undermine control mechanisms that regulate MMPs and A disintegrin and metalloproteases (ADAMs) under normal physiological conditions. For instance, in gastric cancer, miR-139 is down-regulated in CAF-derived exosomes, while its target MMP-11 is enriched, leading to heightened invasive and metastatic abilities of recipient cancer cells [84]. Another study reported the induction of the CAF phenotype in breast, lung, head and neck, and renal cancer cell lines through the knockdown of TIMP, a modulator of ECM integrity via post-translational regulation of MMPs and ADAMs. TIMP-less fibroblasts released exosomes rich in ADAM10, and were able to accelerate tumor growth in vivo in all the mentioned cancers except for renal cancer, which lacks a major stromal component [145].

### 5.3. Other Cargo (lncRNA)

A limited number of studies showcased that CAF-derived EVs may also confer pro-tumorigenic properties, including proliferation, chemoresistance, stemness, and metabolic reprogramming to malignant cells through the transfer of lncRNAs [78,82,147]. Recent findings showed lncRNA small nucleolar RNA host gene 3 (SNHG3) functioned as an endogenous sponge for miR-330, thereby modulating the expression of pyruvate kinase M1/2 (PKM), which led to a down-regulation in cancer cell mitochondrial activity and promoted cancer growth and glycolysis [78].

Overall, further investigations are needed to understand the bidirectional signaling and molecular cargo transfer network established through EVs between cancer cells and CAFs within the TME.

**Table 2 ijms-21-06837-t002:** Summary of effects of cancer associated fibroblast-derived extracellular vesicles on cancer cells.

Author (Year)	Type of Cancer	EV Type; Size *	Isolation Method	Target Cargo	Main Results
**MicroRNA**					
Wang (2020) [142]	Breast	sEVs (exosomes); 30–120 nm	Centrifugation	miR-181d-5p	Delivery of miR-181d-5p via CAF sEVs promoted cell proliferation, invasion, migration, EMT, and inhibited cell apoptosis, through repression of CDX2 and HOXA5; in vivo study showed CAF mediated delivery of miR-181d-5p mimic increased tumor growth rate, volume, and weight.
Kim (2020) [79]	Breast	EVs; size NR	Precipitation via ExoQuick-TC	miR-4516	Loss of CAF derived EV miR-4516 contributes to the proliferation of triple negative breast cancer cells via repression of FOSL1.
Wang (2019) [85]	HNC	EVs; 30–150 nm	Ultracentrifugation with filtration	miR-3188	Down-regulation of miR-3188 in CAF-derived exosomes promotes HNC cell proliferation and inhibits apoptosis; miR-3188 had little effect on migration or invasion; loss of miR-3188 in exosomes leads to BCL2 de-repression in recipient cells.
Wang (2019) [154]	Osteosarcoma	sEVs (exosomes); 30–150 nm	Ultracentrifugation with filtration	miR-1228	miR-1228 expression was significantly increased in CAFs, their secreted exosomes, and recipient osteosarcoma cells; leading to a down-regulation of SCAI mRNA and protein expression in osteosarcoma cells, promoting cell migration and invasion.
Sun (2019) [81]	OSCC	EVs; size NR	Precipitation via Hieff Quick Exosome Isolation Kit	miR-382-5p	CAF-rich OSCC tumor tissues were associated with lymph node metastasis and higher TMN staging; miR-382-5p transfer from CAFs to OSCC cells via exosomes induced cell migration and invasion.
Hu (2019) [86]	CRC	sEVs (exosomes); 50–100 nm	Ultracentrifugation; precipitation via ExoQuick Exosome kit	miR-92a-3p	CAF-derived exosomes promote growth, invasion, metastasis, and chemotherapy (5-FU/L-OHP) resistance in CRC; CAF-derived exosome mediated transfer of miR-92a-3p activates Wnt/β-catenin pathway and inhibited mitochondrial apoptosis through FBXW7 and MOAP1 inhibition.
Guo (2019) [148]	Ovarian	sEVs (exosomes); 50–100 nm	Ultracentrifugation with filtration	miR-98-5p	CAF-derived exosomes deliver miR-98-5p to ovarian cancer cells, contributing to cisplatin resistance by promoting cell proliferation and colony formation, and inhibiting cell apoptosis; CAF-derived exosome miR-98-5p targets CDKN1A, leading to its down-regulation in recipient ovarian cancer cells.
Li (2018) [83]	OSCC	sEVs (exosomes); 40–120 nm	Ultracentrifugation; sucrose cushion centrifugation	miR-34a-5p	miR-34a-5p is down-regulated in CAF exosomes; miR-34a-5p overexpression suppressed cell proliferation, colony formation, migration, invasion, and reduced the weight of tumor nodules; overexpression of AXL, a target of miR-34a-5p, abolished the inhibitory effects of the miRNA through the AKT/GSK-3B/B-catenin signaling pathway, and subsequent up-regulation of SNAIL and activation of MMP-2 and MMP-9.
Li (2018) [88]	Endometrial	sEVs (exosomes); ~70–120 nm	Ultracentrifugation	miR-148b	CAFs promoted metastasis to lungs in vivo; Low levels of miR-148b within CAF-derived exosomes promote endometrial cancer cell migration and invasion; loss of miR-148 results in de-repression of its target DNMT1.
Zhang (2017) [155]	HCC	EVs; size NR	Life Technologies exosome precipitation kit	miR-320a	Decreased levels of miR-320a in CAF-derived exosomes promote cell proliferation, migration, and metastasis; loss of miR-320a leads to de-repression of its target PBX3, activating MAPK pathway and up-regulation of CDK2 and MMP2; in vivo, miR-320a overexpression suppresses tumor growth and metastasis.
Li (2017) [156]	CCA	EVs; mode 93.1 nm, 94.1 nm	Ultracentrifugation	miR-195	miR-195 is down-regulated in CCA cells and CAFs; overexpression of miR-195 in CAF-derived exosomes inhibits growth and invasion of CCA cells; in vivo, CAF-derived exosomes overexpressing miR-195 inhibit CCA tumor growth and increased overall survival.
Khazaei (2017) [89]	ESCC	EVs; size NR	Ultracentrifugation	miR-451	miR-451 was found to be down-regulated in ESCC tumor tissue, but up-regulated in ESCC patient serum and serum exosomes; miR-451 overexpression in fibroblasts increased migration and MIF levels in ESCC cells.
Donnarumma (2017) [143]	Breast	EVs; size NR	Precipitation via ExoQuick-TC	miR-21, miR-378e, miR-231	CAF-derived exosomes increase mammosphere formation, stemness, and EMT; miR-21, -378e, and -231 are up-regulated in CAF-derived exosomes; these miRNAs promote stemness and EMT in breast cells when overexpressed in CAF- derived exosomes and normal fibroblast-derived exosomes.
Bhome (2017) [12]	CRC	sEVs (exosomes); 40–120 nm	Ultracentrifugation with filtration	miR-21	CRC CAF-derived exosome signature consists of miR-329, -181a, -199b, -382, -215, and -21; miR-21 up-regulation; CAFs overexpressing miR-21 promoted tumor liver metastasis in vivo.
Au Yeung (2016) [150]	Ovarian	sEVs (exosomes); ~70–130 nm	Ultracentrifugation with filtration	miR-21	CAF-derived exosomal transfer of miR-21 to ovarian cancer cells suppressed apoptosis and conferred increased chemoresistance (paclitaxel); miR-21 down-regulated its direct target APAF1, in recipient cancer cells.
Shah (2015) [144]	Breast	sEVs (exosomes); 30–100 nm	Ultracentrifugation	miR-221; miR-222	Hyperactive MAPK signaling signature secreted in greater amounts by basal breast cancer CAFs compared to luminal breast cancer CAFs; signature includes overexpression of miR-221/222 in CAF- derived exosomes; CAF23_BAS_-exosomes caused up-regulation of hyperactive MAPK-mediated ER repression in ER+ breast cancer cells.
Josson (2015) [152]	Prostate	EVs; size NR	Precipitation via ExoQuick-TC	miR-409-5p; miR-409-3p	miR-409 expression correlated with higher Gleason score in prostatic and bone tissue; miR-409 overexpression in normal fibroblasts induces CAF-like phenotype; CAF-derived EV mediated transfer of miR-409 to prostate epithelium resulted in repression of tumor suppressors (RSU1, STAG2), promotion of EMT and cell growth, and increased survival.
**Protein**					
Zhao (2020) [140]	ECCS	sEVs (exosomes); mean 50–150 nm	Ultracentrifugation	SHH	CAF-derived exosomes, rich in SHH, increased the expression of PTCH1, SMO, and GLI1 in ESCC cells, indicating activation of the SHH signaling pathway, resulting in enhanced migratory and proliferative abilities of ESCC cells. CAF-derived exosomes increased tumor weight and volume in vivo, this effect could be reversed by cyclopamine.
You (2019) [80]	Lung	sEVs (exosomes); 30–100 nm	Precipitation via ExoQuick-TC	Snail1	CAF-derived exosomes promote EMT of epithelial lung cancer cells, through Snail1 signaling.
Xu (2019) [84]	Gastric	EVs; size NR	Exosome isolation kit (not specified)	MMP11, miR-139	In CAF-derived exosomes, miR-139 is down-regulated while its target MM-P11, is up-regulated; resulting in increased growth, invasion, and metastasis of gastric cancer cells in vitro and in vivo.
Principe (2018) [141]	OTSCC	sEVs; 40–100 nm	Ultracentrifugation	MFAP5	Identified 415 proteins unique to CAF-secretome; MFAP5 enriched in CAF exosome, promotes OTSCC cell growth and migration through activation of MAPK/AKT pathways.
Miki (2018) [87]	Gastric	sEV (exosomes); mean 100–200 nm	Ultracentrifugation with filtration	CD9	CAF-derived exosomes taken up by scirrhous-type of gastric cancer but not in the other types; CD9-positive CAF-derived exosomes promoted migration and invasion in scirrhous-type gastric cancer cells through MMP-2 activation.
Li (2017) [149]	Ovarian	sEVs (exosomes); 30–150 nm	Ultracentrifugation	TGFβ1	TGFβ1 is significantly up-regulated in exosomes from CAFs in ovarian cancers with omental metastasis; TGFβ1 promotes EMT in ovarian cancer through activation of SMAD2/3 signaling.
Chen (2017) [13]	Breast	sEVs (exosomes); majority 80–300 nm	Ultracentrifugation	Wnt10b	Loss of p85a can activate fibroblasts, express high levels of Wnt10b; p85a^-/-^ fibroblast-derived exosomes deliver Wnt10b to breast cancer cells, promoting EMT and inducing Wnt/B-catenin signaling.
Leca (2016) [90]	Pancreatic	EVs- size NR	Ultracentrifugation	ADXA6; ADXA6/LRP1, TSP1 complex	Under pathophysiological conditions (co-culture with macrophages, hypoxia, and lipid starvation), ADXA6/LRP1/TSP1 protein complex promotes PDA cancer aggressiveness; efficient uptake of CAF-derived EVs requires ADXA6; ADXA6+ EVs from CAFs improved recipient cancer cells’ migration; PDA patients were found to have elevated levels ADXA6+ EV in serum.
Santi (2015) [151]	Prostate and melanoma	sEVs (exosomes); mean 3.75 nm EVs (ectosomes); mean 953.3 nm	Ultracentrifugation (exosomes); centrifugation (ectosomes)	Lipids, proteins	Proteins transferred from CAFs to tumor cells increased their proliferation and promoted reverse-Warburg phenotype; ectosome-mediated transfer of proteins had a more pronounced impact on recipient tumor cell proliferation than exosomes.
Shimoda (2014) [145]	Breast	sEVs (exosomes); 40–110 nm	Ultracentrifugation with filtration	ADAM10	TIMP knockout induces CAF-like phenotype in fibroblasts; exosomes derived from TIMPless fibroblasts are rich in ADAM10; exosome delivered ADAM10 promotes cell motility and activates Notch1 and RhoA signaling in breast cancer cells.
Luga (2012) [146]	Breast	sEVs (exosomes); 30–100 nm	Ultracentrifugation	CD81	CD81+ CAF-derived exosomes promote breast cancer cell protrusions and motility, and tumor metastasis in vivo; breast cancer cell produced Wnt11 interacts with exosomes received from CAFs to activate PCP signaling.
Castellana (2009) [153]	Prostate	EVs (microvesicles); size NR	Centrifugation	CX3CL1	Tumor cell MVs promote MMP-9 and ERK1/2 phosphorylation in fibroblasts to induce chemoresistance and migration; in turn activated fibroblasts secrete MVs carrying CX3CL1 that more strongly promote migration/invasion of cancer cells expressing CX3CR1.
**Other cargo**					
Li (2020) [78]	Breast	sEVs (exosomes); 40–100 nm	Ultracentrifugation	SNHG3	CAF-derived exosomes deliver SNHG3 to breast cancer cells, serving as a molecular sponge suppressing miR-330-5p expression; promote breast cancer glycolysis and growth through PKM (miR-330-5p target) modulation.
Yan (2019) [147]	Bladder	sEVs (exosomes); 50–180 nm	Ultracentrifugation	LINC00355	CAF-derived exosomes rich with LINC00355, promote bladder cancer cell proliferation and invasion.
Ren (2018) [82]	CRC	sEVs (exosomes); 10–180 nm	Ultracentrifugation with filtration	lncRNA H19	CAFs promote stemness and chemotherapy (oxaliplatin) resistance in CRC cells through the transfer of exosomal lncRNA H19; lncRNA H19 acts as an endogenous sponge for miR-141, de-repressing the B-catenin pathway; in vivo, CAF-derived exosomes promoted tumor growth.

Legend: (*) Size is reported as range unless otherwise stated.ADAM10, A disintegrin and metalloproteases 10; ADXA6, annexin A6; AKT, protein kinase B; APAF1, apoptotic protease activating factor 1; AXL, AXL receptor tyrosine kinase; BCL2, B-cell lymphoma 2; CAC, colitis-associated cancer; CAF, cancer-associated fibroblast; CCA, cholangiocarcinoma; CDK2, cyclin dependent kinase 2; CDKNIA, cyclin-dependent kinase inhibitor 1A; CDX2, caudal-related homeobox 2; CRC, colorectal cancer; CX3CL1, C-X3-C motif chemokine ligand 1; DNMT1, DNA methyltransferase 1; EMT, epithelial-mesenchymal transition; ER, estrogen-receptor; ERK, extracellular-signal-regulated kinase; ESCC, esophageal squamous cell carcinoma; EVs, extracellular vesicles; FBXW7, F-box and WD repeat domain containing 7; FOSLI, Fos-related antigen 1; GLI1, glioma-associated oncogene homolog 1; GSK-3B, glycogen synthase kinase 3 beta; HCC, hepatocellular carcinoma; HNC, head and neck cancer; HOXA5, Homeobox A5; lncRNA, long non-coding RNA; LRP1, LDL receptor related protein 1; MAPK/AKT, Mitogen-activated protein kinase/protein kinase B; MFAP5, Microfibril associated protein 5; MIF, macrophage migration inhibitory factor; MMP, matrix metalloproteinase; MOAP1, modulator of apoptosis 1; MVs, microvesicles; NOTCH1, Notch homolog 1, translocation associated; NR, not reported; OSCC, oral squamous cell carcinoma; OTSCC, oral tongue squamous cell carcinoma; PBX3, PBX homeobox 3; PCP, planar cell polarity; PDA, pancreatic ductal adenocarcinoma; PKM, pyruvate kinase M1/2; PTCH1, patched 1; RhoA, Ras homolog family member A; RSUI, Ras suppressor protein 1; SCAI, suppressor of cancer cell invasion; sEVs, small extracellular vesciles; SHH, Sonic Hedgehog; SMAD2/3, SMAD family member 2/3; SMO, smoothened; SNAI1, snail family transcriptional repressor 1; SNHG3, small nucleolar RNA host gene 3; STAG2, stromal antigen 2; TIMP, tissue inhibitor of metalloproteinases; TGFβ, transforming growth factor beta; TSP1, thrombospondin-1; Wnt10b, Wnt family member 10b.

## 6. CAF-Derived EVs Contents in Therapy Resistance

CAF-derived EVs introduce a novel mechanism of acquired resistance to anticancer treatment (i.e., chemotherapy or hormonal therapy). Therapy resistance can occur through EVs cargos delivered from CAFs to other cells in the TME and may involve the regulation of several signaling pathways by these cargos [159].

CAFs demonstrate a high degree of resistance to chemotherapeutics and can transfer this resistance to neighboring cancer cells via EV miRNAs. For instance, exosomes derived from gemcitabine-resistant CAFs contain high levels of miR-106a that, when taken up by recipient pancreatic cancer cells, promotes chemoresistance through the inhibition of the tumor suppressor TP53INP1 [160]. In ovarian cancer, CAF-derived exosomes express high levels of miR-98-5p that is transmitted to cells of the TME, leading to the development of cisplatin resistance by cancer cells through direct inhibition of cyclin-dependent kinase inhibitor 1A (CDKN1A, p21) [148]. Further, miR-21 is secreted via EVs from cancer-associated adipocytes and CAFs, leading to inhibition of APAF1 in recipient cells of the TME [150]. MiR-21 mediated repression of APAF1 inhibits cell cycle arrest and promotes resistance to chemotherapy [150]. As an additional example of how CAF-cancer cell cross-talk can increase chemoresistance, activation of the ubiquitin-specific protease 7 (USP7)/heterogeneous nuclear ribonucleoprotein A1 (hnRNPA1) axis in gastric cancer cells exposed to paclitaxel and cisplatin increases the secretion of miR-522 in CAF-derived exosomes, leading to ferroptosis suppression in tumor cells and culminating in increased chemoresistance [161].

In addition to miRNAs, CAF-derived EV lncRNAs have also been linked to increased chemoresistance. For instance, colorectal cancer-associated lncRNA (CCAL) is an oncogenic lncRNA highly expressed in CAFs associated with colorectal cancers, and its transference to the TME via exosomes is associated with chemoresistance [14]. Similarly, lncRNA H19 is significantly up-regulated in CAF-derived exosomes in colorectal cancer and acts as a competing endogenous RNA sponge to tumor suppressive miR-141 [82]. Inhibition of miR-141 promotes the stemness of cancer cells and leads to the activation of the Wnt/β-catenin signaling pathway [82]. Transmission of exosomal H19 from CAFs to the neighboring cells is strongly associated with tumor development and resistance to oxaliplatin [82].

Cancer stem cells (CSCs) have also been shown to exhibit a high degree of resistance to common cancer treatments [162]. Interestingly, CAF-derived exosomes were found to increase the proportion of CSCs and induce their tumorigenic capability in colorectal cancer patient-derived xenografts, which led to the development of 5-fluorouracil and oxaliplatin resistance [163]. Inhibition of CAF-derived EVs decreased the proportion of CSCs and abrogated their tumorigenic potential [163]. In agreement, CAF-derived exosomes were found to transfer Wnt proteins to colorectal cancer cells, thereby reprogramming them into CSCs and increasing their resistance to chemotherapy [164]. In addition, incubation of luminal breast cancer cells exposed to hormonal therapy with miR-221-containing CAF-derived exosomes resulted in the activation of the ER/Notch feed-forward loop, the generation of CD133 CSCs with low expression levels of estrogen receptor alpha, and consequently, the development of de novo hormonal therapy resistance [165]. Reciprocally, CSC-derived EVs up-regulate the β-catenin/mTOR/STAT3 pathway and increase mRNA and protein levels of TGF-β1 [99]. These EVs can transform normal fibroblasts into CAFs with enhanced oncogenic potential; in turn, these CAFs then increase chemoresistance in oral cancer cells [99]. Although these results are intriguing, additional studies are required to fully elucidate the mechanisms underlying CSC-CAF interactions and their relationship to drug resistance.

## 7. EVs and Pre-Metastatic Niche Formation

Evidence supports the concept that primary pro-metastatic cancer cells can promote metastasis by inducing changes in the microenvironment of distant organ sites, namely, at pre-metastatic niches [166]. In this context, EV exchange between primary tumor cells and the pre-metastatic niche was shown to play an important role as a communication medium, being able to influence several processes in the pre-metastatic site (e.g., angiogenesis, ECM remodeling, immune response, and inflammation) [167]. For instance, in vivo experiments showed that EVs that originated from nasopharyngeal cancer carrying LMP1 were able to increase the expressions of fibronectin, S100A8, and VEGFR1 in lung and liver tissues [127]. Additionally, there is evidence that EVs might mediate organ-specific tropism of metastasis. The pattern of integrin expression in tumor-derived exosomes was associated with adhesion to specific ECM molecules and cell types; for example, ITGα6β4 and ITGα6β1 expression lead to lung tropism, while ITGαvβ5 expression mediates liver tropism [168].

Pre-metastatic niche formation is frequently associated with the activation of fibroblasts in a distant site by tumor cell-derived EVs. In this regard, metastatic breast cancer cells release significantly higher amounts of transglutaminase-2 and fibrillar fibronectin, which is associated with an enhanced growth-supportive phenotype in lung fibroblasts and metastatic niche formation in lung tissues [169]. Triple-negative breast cancer-derived EVs are also proposed to mediate pre-metastatic changes in ECM and soluble components of lung tissue, including enhanced expression of fibronectin and periostin in lung fibroblasts [170]. Exosomal miR-155 and miR-219 secreted by melanoma cells are able to induce metabolic reprograming in normal fibroblasts by favoring glycolysis over oxidative phosphorylation, leading to an acidified extracellular pH, which is associated with pre-metastatic niche formation [108]. Furthermore, breast cancer cell-derived EVs containing miR-122 are understood to down-regulate glucose consumption in lung fibroblasts, astrocytes, and neurons, with miR-122 inhibition restoring glucose uptake and decreasing the incidence of metastasis in vivo [171].

Cancer cell-derived EVs are also associated with pro-inflammatory profiles in pre-metastatic niche fibroblasts. Metastatic hepatocellular carcinoma cells secrete miR-1247-3p via exosomes, leading to activation of the β1-integrin–NF-κB pathway in fibroblasts and promoting a pro-inflammatory profile in these fibroblasts, which was associated with pre-metastatic niche formation and lung metastasis [111]. EVs derived from colorectal cancer cells containing integrin beta-like 1 (ITGβL1) stimulate the activation of resident fibroblasts in distant organs via TNFAIP3-mediated NF-κB pathway. As a result, the activated fibroblasts secrete pro-inflammatory cytokines and promote metastatic cancer growth [172]. Furthermore, EVs secreted by metastatic melanoma cells induce a pro-inflammatory profile in lung fibroblasts without any differences in wound healing functions, while also triggering pro-inflammatory signaling in astrocytes [138].

Fewer studies have assessed the ability of CAF-derived EVs to promote pre-metastatic niche formation. In salivary adenoid cystic carcinoma, CAF-derived EVs were able to induce pre-metastatic niche formation in lung tissue, including enhanced matrix remodeling, periostin expression, and lung fibroblast activation [173]. However, the role of CAF-derived EVs in pre-metastatic niche formation remains to be elucidated in further studies.

## 8. Conclusions

Extracellular vesicles are proven to play important roles as mediators of intercellular communication, and therefore, have been targets of increasing interest in cancer research. EVs affect different molecular pathways in the cross-talk between cancer cells, but also between these cells and the stromal components of the TME. Different cargos contained in EVs secreted by cancer cells are able to influence TME cells to exert pro-tumorigenic functions, which include induction of CAF, pro-inflammatory, and pro-angiogenic phenotypes. The activation of a CAF phenotype in stromal cells can alter the contents of their secreted EVs. CAF-derived EVs have been shown to promote tumor progression by influencing cancer cells to develop more aggressive characteristics, including increased growth, migration, invasion, metastasis, and therapy resistance. Additionally, evidence shows that EVs derived from cancer cells and CAFs are able to influence the microenvironments at distant sites and promote pre-metastatic niche formation. This line of research inquiry holds promise for clinical utility via possible identification of novel biomarkers and therapeutic targets, and the possible use of EVs as a vector for delivery of therapeutic agents [155,156,174].

It is important to emphasize that the majority of evidence regarding EVs’ role in tumorigenesis has been derived from in vitro and animal model studies, with limited data regarding the utility of these findings for clinical applications. Although EV-based research seems promising to clarify the molecular mechanisms involved in the cross-talk between CAFs and cancer cells during tumorigenesis, new tools and/or research methods need to be developed to apply the findings in clinical settings.

## Figures and Tables

**Figure 1 ijms-21-06837-f001:**
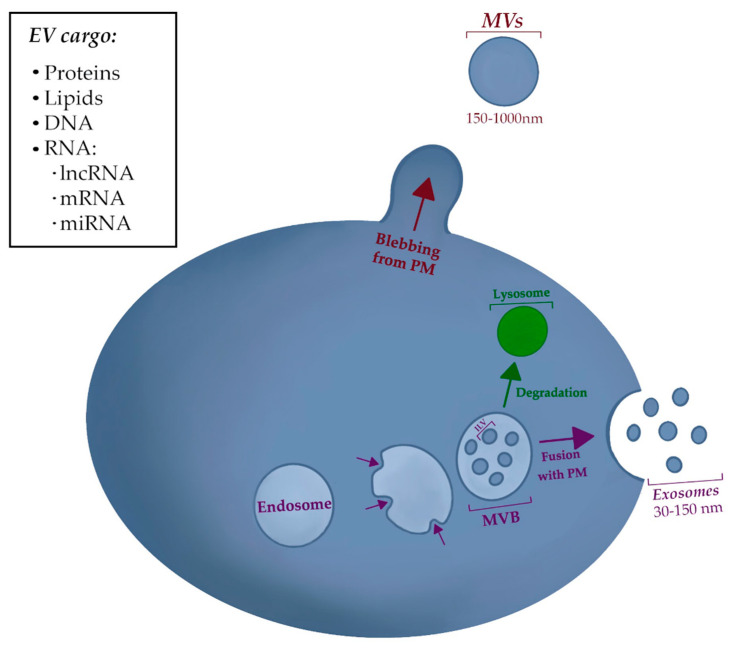
Schematic representation of microvesicles (MVs) and exosomes’ biogenesis and release by eukaryotic cells. MVs are formed after an outward blebbing of the plasma membrane (PM) and are usually sized between 150 and 1000 nm. Exosome (30–150 nm) formation begins in late endosomes that maturate into multivesicular bodies (MVB), including the formation of intervesicular bodies (ILVs) through the inward budding of the MVs limiting membrane. MVB can fuse with the plasma membrane and release the ILVs into the extracellular space as exosomes. Alternatively, MVB can suffer degradation by fusing with lysosomes. EVs can contain numerous biomolecules, including protein, lipids, DNA, and RNA.

**Figure 2 ijms-21-06837-f002:**
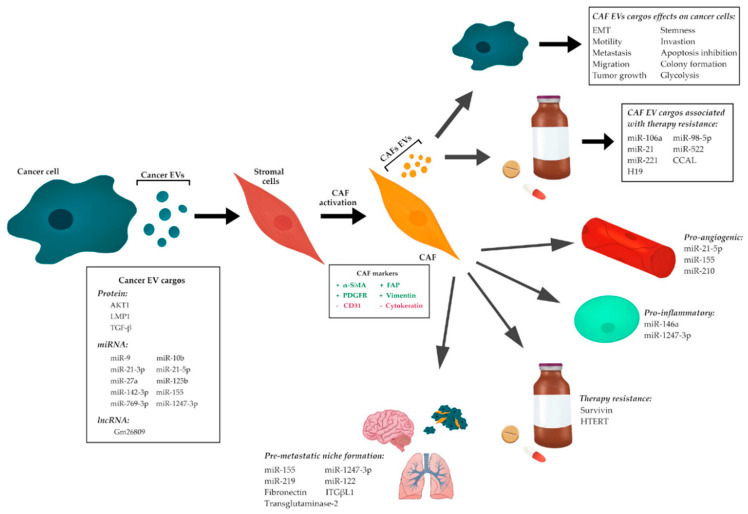
Summary of the extra-cellular vesicle (EVs)-mediated cross-talk between cancer cells and cancer-associated fibroblasts (CAFs). Cancer cells can influence stromal cells to activate a CAF phenotype through the release of EVs, which carry several cargos, including proteins, micro-RNAs (miRNA), and long noncoding-RNA (lncRNA). Specific cancer cells-derived EV cargos can also influence a pro-angiogenic or pro-inflammatory phenotype in CAF, and the induction of therapy resistance and pre-metastatic niche formation. At the same time, CAF-derived EVs cargos can influence cancer cells to increase epithelial-to-mesenchymal transition (EMT), growth, invasion, metastasis, motility, stemness, colony formation, apoptosis inhibition, glycolysis, and therapy resistance.

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
