# Peer review of "The Role of Cancer-Associated Fibroblasts and Extracellular Vesicles in Tumorigenesis"

_ijms, 2020, doi:10.3390/ijms21186837_

Round 1
Reviewer 1 Report
The manuscript by Shoucair et al. entitled “The role of cancer-associated fibroblasts and extracellular vesicles in tumorigenesis” reviews numerous studies implicating the roles of cancer-associated fibroblasts (CAFs) in promoting tumorigenesis, in particular through secretion of extracellular vesicles that mediate crosstalk in a tumor microenvironment (TME). Although not conceptually novel, overall the review is very thorough and contains detail summarizing accumulating evidence in this field. However, several points should be addressed prior to publication:
Major points:
- In general, the text is very detailed and lengthy, and thus the flow of the article is difficult to follow. There is a significant body of information that is extraneous to the concept of the review article, including several pages of discussion of endothelial cell or immune cell contribution to the TME that could be condensed or deleted altogether.
- Subtitles are recommended to organize large sections of discussion.
- Much of the detail in Section 4 could be summarized in a table.
- Detail in section 5 and Table 1 is often redundant.
- The authors should refer to secreted vesicles as “extracellular vesicles” according to Minimal information for studies of extracellular vesicles 2018 (MISEV2018) standards (PMID: 30637094) unless the origin of vesicle biogenesis is well characterized in the primary studies cited.
- An interplay between endosomal-derived vesicles and autophagy has been recently recognized, with emerging evidence also supporting the importance of autophagy in CAF activation. Can the authors expand on these cellular mechanisms with regard to fibroblasts in the TME?
Minor points:
- How is Table 1 organized? Studies pertaining to the same target cargo should be grouped together.
- Font sizes should be increased in Figure 2 for better readability.
Author Response
Point 1: "The manuscript by Shoucair et al. entitled "The role of cancer-associated fibroblasts and extracellular vesicles in tumorigenesis" reviews numerous studies implicating the roles of cancer-associated fibroblasts (CAFs) in promoting tumorigenesis, in particular through secretion of extracellular vesicles that mediate cross-talk in a tumor microenvironment (TME). Although not conceptually novel, overall the review is very thorough and contains detail summarizing accumulating evidence in this field. However, several points should be addressed prior to publication:"
Response: We thank the reviewer for the insightful critiques regarding our manuscript. Their suggestions helped us to improve the overall quality and clarity of the revised manuscript and we appreciate their suggestions.
Point 2: "In general, the text is very detailed and lengthy, and thus the flow of the article is difficult to follow. There is a significant body of information that is extraneous to the concept of the review article, including several pages of discussion of endothelial cell or immune cell contribution to the TME that could be condensed or deleted altogether."
Response: As suggested by the reviewer we removed sub-sections regarding extracellular matrix, blood and lymphatic vessels, and inflammatory and immune cells in section 2. As a result, the section was renamed "cancer-associated fibroblasts (CAFs)" and only information regarding CAFs interaction with other cells of the TME was maintained.
Point 3: "Subtitles are recommended to organize large sections of discussion."
Response: Thank you for your suggestion. Sub-headings were included in section 4 and 5 to better organization; all contents were sub-classified according to EV cargo type (miRNA, protein, and others).
Point 4: "Much of the detail in Section 4 could be summarized in a table."
Response: A new table was included "Table 1" and information on section 4 was summarized (pages 5-15).
Point 5: "Detail in section 5 and Table 1 is often redundant."
Response: Redundant information in section 5 was removed according to the reviewer's comment.
Point 6: " The authors should refer to secreted vesicles as "extracellular vesicles" according to Minimal information for studies of extracellular vesicles 2018 (MISEV2018) standards (PMID: 30637094) unless the origin of vesicle biogenesis is well characterized in the primary studies cited."
Response: The manuscript was revised, and all generic terms were modified to extracellular vesicles ("EVs").
Point 7: "An interplay between endosomal-derived vesicles and autophagy has been recently recognized, with emerging evidence also supporting the importance of autophagy in CAF activation. Can the authors expand on these cellular mechanisms with regard to fibroblasts in the TME?"
Response: We have added a paragraph regarding the possible role of autophagy in CAFs activation and consequences in tumorigenesis.
Below are the items altered in the manuscript:
- Section 2, cancer-associated fibroblasts, page 02, line 79-88: "Another important process for CAF differentiation is the metabolic reprogramming, which is linked to several processes, including cancer-cells induced oxidative stress, which modifies mitochondrial function resulting in higher glucose uptake and reactive oxygen species, culminating in CAF differentiation [31]. Curiously, studies show that CAFs can also undergo autophagy under the influence of the increased demands for energy from other cells of the TME [32]. For instance, cancer cells induce oxidative stress in CAFs, which results in autophagy and secretion of nutrients that enhance an aggressive phenotype in these cancer cells [33]. In this context, cells from the TME secrete high-energy metabolites (e.g., lactate and ketone) and induce a metabolic reprogramming in CAFs, including autophagy and aerobic glycolysis, which influence several processes during tumorigenesis, including inflammation, angiogenesis, and tumor growth [34, 35]."
Point 8: "How is Table 1 organized? Studies pertaining to the same target cargo should be grouped together."
Response: Thank you for your suggestion. Table 1 was re-organized according to target cargo.
Point 9: "Font sizes should be increased in Figure 2 for better readability."
Response: According to the reviewer's comment we modified the page layout for figure 2 for landscape.
Reviewer 2 Report
This review by Shoucair et al provides a discussion about the role of extracellular vesicles in the cross-talk between cancer cells and carcinoma associated fibroblasts in cancer progression.
This work provides a comprehensive review on several cancer cells-derived EV on CAF activation and function as well as those CAF-derived on cancer cell biology.
This is a well-presented review that focus on CAF biology and EV,
Given that EV englobes different types (exosomes, microvesicles and apoptotic bodies), it will be interesting to have a small discussion about:
- When does a CAF use exosomes vs microvesicles or both?
- What are the mechanisms that govern the production/use of either?
- Are these mechanisms targetable?
- How about in the clinic, can any of these be used to assess diagnosis/prognosis?
Perhaps another aspect that needs some discussion is about the technical aspects associated with the study of EV. The isolation of EV and exosomes requires a substantial amount of material (large number of cells). Given that CAF can only be cultured for short period of time because they only retain their pro-tumorigenicity for a few passages, it present challenges. Especially when using human-derived CAF.
While many in vitro experimental studies have shown the role of EV on CAF biology it is still unclear the contribution in the TME of cancer patients. There is no clear clinical evidence, therefore new tools are needed to assess the translational relevance.
The authors mentioned that EV can also affect other cell types in the TME but offer no discussion about the cross-talk between CAF and immune-inflammatory cells or other cells.
It is well recognized the effects of immunosuppressive and pro-inflammatory role of CAF in cancer. A brief discussion should be included.
One final aspect is that EV can be beneficial and inhibit tumor progression via antigen presentation to immune cells, for example rat glioblastoma-derived exosomes can induce anticancer response in combination with galactosylceramide (Liu HY, Cancer Letters 2017). Therefore, more studies are needed to explore the utility of using some of these potential anti-tumor EV effects.
Author Response
Point 1: "This review by Shoucair et al provides a discussion about the role of extracellular vesicles in the cross-talk between cancer cells and carcinoma associated fibroblasts in cancer progression.
This work provides a comprehensive review on several cancer cells-derived EV on CAF activation and function as well as those CAF-derived on cancer cell biology.
This is a well-presented review that focus on CAF biology and EV"
Response: Thank you for the positive feedback regarding our review. We would like to thank the reviewer for the time and effort in critically revising our manuscript, which helped us to improve the overall quality of its contents.
Point 2: "Given that EV englobes different types (exosomes, microvesicles and apoptotic bodies), it will be interesting to have a small discussion about:
When does a CAF use exosomes vs microvesicles or both?
What are the mechanisms that govern the production/use of either?
Are these mechanisms targetable?
How about in the clinic, can any of these be used to assess diagnosis/prognosis?"
Response: a small discussion addressing these questions has been added.
Below are the items altered in the manuscript:
- Section 3, extracellular vesicles, page 05, line 203-213: "Damaged and diseased cells, including cancer, have been shown to shed higher amounts of EVs compared to their healthy counterparts. This heightened production may be linked to the extensive metabolic reprogramming that cancer cells undergo [76]. The mechanisms regulating EV production are unclear, however EV formation and release have been inhibited by targeting biological molecules involved in EV trafficking (Calpeptin, Manumycin A, Y27632) or lipid metabolism (D-pantethine, imipramine, GW4869) [77]. In regards to fibroblasts, GW4869 which targets nSMase to inhibit exosome generation and release, has been extensively used in CAF-EV research [13, 78-83].
Several studies [79, 81, 84-90] have included an investigation into clinical translations of their findings where CAF-EV cargo expression in tissue or blood based biopsies were found to be linked to clinical features such as overall survival [88], lymph node metastasis [87], metastasis [84], or poor prognosis [79]."
Point 3: "Perhaps another aspect that needs some discussion is about the technical aspects associated with the study of EV. The isolation of EV and exosomes requires a substantial amount of material (large number of cells). Given that CAF can only be cultured for short period of time because they only retain their pro-tumorigenicity for a few passages, it present challenges. Especially when using human-derived CAF."
Response: The manuscript has been revised to include a discussion of EV isolation techniques to section 3 ‘Extracellular vesicles (EVs)’.
Below are the items altered in the manuscript:
- Section 3, extracellular vesicles, pages 04-05, line 192-202: "There are several techniques employed currently to isolate EVs. Differential ultracentrifugation (DU) is widely used as a low-cost, high-throughput method to isolate EVs from large sample sizes. DU is often combined with other purification techniques such as ultrafiltration or density gradient centrifugation to yield increased particle purity. Size-exclusion chromatography yields EV pellets of high purity but dilutes samples, which need to then be re-concentrated. Immuno-affinity and microfluidics are techniques based on EV characteristics such as surface markers (e.g. CD63) but there is currently no marker that can accurately discern between various EV subtypes. Polymer based precipitation capture EVs in polymer nets based on size using simple centrifugation making it useful for clinical usage. EV samples are further characterized commonly through transmission electron microscopy (TEM), nanoparticle tracking analysis, western blots for common markers (TSG101, Alix, CD63, CD81, Floatillin), and flow-cytometry [74, 75]."
Point 4: "While many in vitro experimental studies have shown the role of EV on CAF biology it is still unclear the contribution in the TME of cancer patients. There is no clear clinical evidence, therefore new tools are needed to assess the translational relevance."
Response: We added a small discussion about the limitations of current evidence regarding EVs research and application in clinical settings.
Below are the items altered in the manuscript:
- Conclusions, pages 22-23, line 534-539: "It is important to emphasize that the majority of evidence regarding EVs role in tumorigenesis is derived from in vitro and animal model studies, with limited data regarding the utility of these findings for clinical application. Although EV-based research seems promising to clarify the molecular mechanisms involved in the cross-talk between CAFs and cancer cells during tumorigenesis, new tools and/or research methods need to be developed to apply the findings in clinical settings."
Point 5: "The authors mentioned that EV can also affect other cell types in the TME but offer no discussion about the cross-talk between CAF and immune-inflammatory cells or other cells.
It is well recognized the effects of immunosuppressive and pro-inflammatory role of CAF in cancer. A brief discussion should be included."
Response: We added a brief discussion regarding the interaction of CAFs with immune and inflammatory cells, focusing on immunosuppressive and pro-inflammatory mechanisms.
Below are the items altered in the manuscript:
- Section 2, cancer-associated fibroblasts, pages 03-04, line 122-149: "Through immunoediting tumor cells and TME cells, including CAFs, can regulate tumor-promoting and protective mechanisms of the immune system during tumorigenesis. Tumor cells are first eliminated by the immune cells (elimination), then tumor cells and immune cells coexist (equilibrium), and lastly tumor cells subclones modify their immunogenicity and evade immune recognition (escape), promoting tumor progression [22]. In this context, CAFs can modulate the tumor immune response (i.e., induction of immunosuppression) to facilitate tumor progression, including interactions with several immune cells, such as macrophages, T cells, dendritic cells, and myeloid cells. For instance, CAFs can recruit monocytes by secreting monocyte chemotactic protein-1 (MCP-1) and stromal cell-derived factor-1 (SDF-1), they are also able to induce transdifferentiation of M1 macrophages to M2 macrophages, which results in immunosuppression and increased cancer cell proliferation [52]. Additionally, CAFs can indirectly reduce T cells activation by secreting cytokines, such as TGF-β, that modulate antigen presentation, which is primarily associated with CAFs effects in dendritic cells [53].
Interestingly, recent evidence suggests that a specific CAF sub-population, namely inflammatory CAFs (iCAFs), is responsible for inducing and maintaining an inflammatory microenvironment through the secretion of several pro-inflammatory cytokines (e.g., IL-6 and CXCL1) [54, 55]. Inflammatory cytokines and chemokines play different roles in tumor progression, including EMT promotion, invasion, and metastasis [56, 57]. Therefore, through the modulation of tumor-associated inflammation, iCAFs also interfere in several processes of tumorigenesis. For example, CAF-derived IL-1β can induce C motif chemokine ligand 22 (CCL22) mRNA expression in oral cancer cell lines through the activation of transcription factor nuclear factor kappa B (NF-κB), which is associated with cell transformation and regulatory T cell infiltration [58]. The interactions between CAFs and the immune and inflammatory network is complex and includes several types of cells (e.g., cytotoxic and helper T cells, macrophages, neutrophils, dendritic cells, natural killer cells, and myeloid-derived suppressor cells) as well as related cytokines, chemokines, and soluble factors (e.g., ILs, TGF-β, and TNF-α). While outside of the main scope of the present review, these interactions between CAF and the immune system and inflammation have been previously reviewed by other authors [22, 23]."
Point 6: "One final aspect is that EV can be beneficial and inhibit tumor progression via antigen presentation to immune cells, for example rat glioblastoma-derived exosomes can induce anticancer response in combination with galactosylceramide (Liu HY, Cancer Letters 2017). Therefore, more studies are needed to explore the utility of using some of these potential anti-tumor EV effects."
Response: The manuscript has been revised to include a discussion of potential anti-tumor effects by CAFs in section 5 ‘CAFs EVs contents in the communication with malignant cells’.
Below are the items altered in the manuscript:
- Section 5, CAFs EVs contents in the communication with malignant cells, page 15, line 363-369: "There is a lack of evidence supporting any anti-tumor effects by CAF-EVs specifically, however some studies have shown possible anti-tumor effects by CAFs [158]. For instance, one study targeted the extensive fibrosis and stromal myofibroblasts associated with pancreatic ductal carcinomas. The deletion of α-SMA myofibroblasts in pancreatic ductal carcinomas mice models enhanced hypoxia, EMT, CSCs, chemoresistance, and reduced animal survival. A decrease in immune surveillance and increased Treg infiltration was also observed, leading the authors to suggest that the fibrosis associated with PDAC is part of a host immune response [159]."
Round 2
Reviewer 1 Report
The authors have addressed most of the major revisions. Several minor revisions are further suggested:
- Additional subtitles could be considered to break up the CAF paragraphs (Section 2).
- Table 1 and 2: Have the studies listing EV type as “Exosomes” confirmed the isolated vesicles to be endosomal-derived vesicles (i.e. bona fide exosomes)? Since the authors added a description of EV isolation techniques in the revised version (lines 192-200), they could also add a column listing the isolation methods of studies cited in Tables 1 and 2.
- Table 1 could also be organized by cargo, instead of citation order.
Author Response
Point 1: Additional subtitles could be considered to break up the CAF paragraphs (Section 2).
Response: As suggested additional subtitles were included in section 2 (page 2, line 68; page 3, line 102).
Point 2: Table 1 and 2: Have the studies listing EV type as “Exosomes” confirmed the isolated vesicles to be endosomal-derived vesicles (i.e. bona fide exosomes)? Since the authors added a description of EV isolation techniques in the revised version (lines 192-200), they could also add a column listing the isolation methods of studies cited in Tables 1 and 2.
Response: An additional column was added to Table 1 and 2 listing the isolation methods of each study. The ‘EV type’ column was modified to include the size of the isolated EVs as reported in the studies. As EV isolation techniques are not perfect, the term small extracellular vesicles (sEVs) was used to recognize that although a sample is likely enriched for exosomes, there may be other EV subtypes within the sample as well. (Table 1 – pages 7-13; Table 2 – pages 18-22).
Point 3: Table 1 could also be organized by cargo, instead of citation order.
Response: Table 1 was modified to be organized by cargo type (pages 7-13).
Reviewer 2 Report
The authors addressed the questions raised during the review process.